# Disseminated Peritoneal Leiomyomatosis—A Challenging Diagnosis-Mimicking Malignancy Scoping Review of the Last 14 Years

**DOI:** 10.3390/biomedicines12081749

**Published:** 2024-08-03

**Authors:** Carmen Elena Bucuri, Razvan Ciortea, Andrei Mihai Malutan, Valentin Oprea, Mihai Toma, Maria Patricia Roman, Cristina Mihaela Ormindean, Ionel Nati, Viorela Suciu, Marina Simon-Dudea, Dan Mihu

**Affiliations:** 12nd Department of Obstetrics and Gynaecology, “Iuliu Hatieganu” University of Medicine and Pharmacy, 400347 Cluj-Napoca, Romania; cbucurie@yahoo.com (C.E.B.); malutan.andrei@gmail.com (A.M.M.); opreacv31@gmail.com (V.O.); mpr1388@gmail.com (M.P.R.); prodan@gmail.com (C.M.O.); nati.ionel@yahoo.com (I.N.); suciuviorela@yahoo.com (V.S.); marina_d3@yahoo.com (M.S.-D.); dan.mihu@yahoo.com (D.M.); 2Clinical Department of Surgery, “Constantin Papilian” Emergency Clinical Military Hospital, 22 G-ral Traian Mosoiu, 400132 Cluj-Napoca, Romania; dr.mtoma.eru@gmail.com

**Keywords:** disseminated peritoneal leiomyomatosis, hormonal theory, myomectomy

## Abstract

Disseminated peritoneal leiomyomatosis (DPL) is a rare condition marked by multiple leiomyomas in the peritoneal cavity, predominantly affecting women of reproductive age. Although typically benign, DPL can present significant diagnostic challenges and, in rare cases, may progress to malignancy. A primary contributing factor to DPL is iatrogenic, particularly due to surgical interventions such as morcellation during myomectomy. This scoping review explores the pathogenesis, epidemiology, diagnosis, and management of DPL, highlighting the crucial role of hormonal influences and iatrogenic factors. Diagnostic methods include computed tomography, ultrasound, magnetic resonance imaging, and histopathological evaluation, which are essential for assessing disease extent and guiding treatment. Management strategies encompass surgical intervention—with a focus on minimizing iatrogenic risks—conservative approaches for asymptomatic patients, and advancements in hormonal treatments. Emphasis is placed on preventing iatrogenic dissemination through refined surgical techniques and patient education. Despite its rarity, with fewer than 200 cases reported globally, understanding DPL’s clinical presentation and iatrogenic origins is vital for optimal patient outcomes. This review underscores the importance of early diagnosis, personalized treatment plans, and ongoing research to address the challenges associated with DPL.

## 1. Introduction

Disseminated peritoneal leiomyomatosis (DPL) is an unusual disease characterized by multiple leiomyomas in the peritoneal cavity [1]. While it is naturally considered gentle and easily curable, DPL often can result in diagnostic difficulties and in irregular cases; the patient may suffer a fatal alteration. Diverse healthcare amenities use other approaches to diagnose and treat DPL. Similarly, several management approaches are utilized to manage DPL to prevent its dissemination or reoccurrence for improved patient results. Although DPL is viewed as a less risky illness, malignant changes have been reported in the review of unusual cases. Therefore, an inclusive evaluation of lesion traits such as rapid growth, scarce imaging foundations, or histopathological results signifying sarcomatous is vital. As Rosati et al. considers, patients suffering from DLP may require better and improved treatment procedures and effective monitoring of the sickness to ensure quality care delivery [2].

## 2. Materials and Methods

This study was conducted as a scoping review to map the existing literature on DPL and identify key concepts, gaps, and evidence related to its pathogenesis, epidemiology, diagnosis, and management. The scoping review methodology was chosen for its suitability in summarizing a broad range of evidence to provide an overview of a complex and emerging topic such as DPL.

In conducting this scoping review, we adhered to the PRISMA (Preferred Reporting Items for Systematic Reviews and Meta-Analyses) guidelines to ensure comprehensive and transparent reporting, under the registration number INPLASY202460092, DOI Number: 10.37766/inplasy2024.6.0092.

The PRISMA 2020 flow diagram was used to document the process of identification, screening, eligibility assessment, and inclusion of sources of evidence in this review. The flow diagram and the detailed process are consistent with PRISMA recommendations for systematic and scoping reviews.

A comprehensive search of the PubMed and Scopus databases was performed to identify all relevant literature published between January 2010 and June 2024. In PubMed, the search query used was “Disseminated Peritoneal Leiomyomatosis” in any field. In Scopus, the same phrase was entered into the title, abstract, and keyword fields. Only English-language articles were included based on the eligibility criteria. The initial search yielded a total of 104 records across the two databases. After removing duplicate articles, the remaining 65 titles and abstracts were screened to determine relevance to the objectives of the review. Articles not focusing on disseminated peritoneal leiomyomatosis or lacking detailed case information were excluded at this stage. The full texts of the 28 potentially eligible articles were then assessed for inclusion. Reference lists of included studies were also manually searched to identify any other relevant publications. This process led to the final inclusion of 21 studies that met the criteria of reporting complete case descriptions diagnosed as disseminated peritoneal leiomyomatosis. Data were then extracted from these 21 articles that comprehensively described the cases (Figure 1).

### 2.1. Peritoneal Leiomyomatosis Pathogenesis, Epidemiology, Differential Diagnosis

Generally, DPL is an unfamiliar sickness that chiefly impacts women during their reproductive stage. Nevertheless, a few incidences have been reported in males. Although DPL is risky, its prevalence and occurrence are yet to be well-known because of its uncommonness. However, it is commonly perceived as a rare condition among leiomyomatous disorders. According to Li and Dai (2020), very few women have been affected by the disease, making it a rare condition [3]. Further, the precise and correct etiology of DPL remains indefinable, and several factors might significantly contribute to its growth and spread in the body. Hormonal influences have been, for the longest time, involved in the pathogenesis of peritoneal leiomyomatosis, especially those happening in the peritoneal cavity, which are the most dangerous.

According to Yang et al. (2021), DPL etiology entails two fundamental theories: the hormone theory and the iatrogenic theory [4]. The iatrogenic theory argues that DPL results from iatrogenic dissemination of uterine leiomyoma and due to morcellation during myomectomy. Commonly, the morcellation of uterine leiomyoma may leave tiny particles of original myomas leading into DPL. Also, Yang et al. (2021) state that the present DPL might be caused by uterine leiomyoma cells seeding after the surgery [4]. Additionally, the hormone theory claims that DPL nodules might originate from the metaplastic changes in stem cells known as mesenchymal into myocytes because of high levels of the stimulation of sex hormones in women. A study by Morgan et al. (2022) shows that many DPL occurrences are usually endogenous, meaning they are identified during pregnancy or when secreting ovarian tumors [5]. 

Other incidences involve exogenous hormones, which happen during hormone replacement therapies. As per the immunohistochemistry staining outcomes, DPL lesions such as estrogen receptors (ERs) and progesterone receptors (PRs) are witnessed more, which shows that sex hormones significantly contribute to nodule progression [4]. Similarly, Thi et al. (2023) state that progesterone and estrogen receptors have been identified in leiomyomatosis tissues, indicating their role in hormonal dysregulation and fostering smooth muscle cell proliferation and peritoneal leiomyoma development [6].

A study by Soni et al. (2020) showed that DPL is mainly one of the three types of uterine smooth muscle tumors with abnormal development series whose histology is the same as that of uterine leiomyoma [1]. Further, the other kinds are intravenous leiomyomatosis and benign metastasizing leiomyoma. According to Yang et al. (2021), to date, fewer than two hundred cases of DPL have been reported globally, the majority being women in their prime [4]. Thus, because of the low incidence rate of DPL, there is inadequate agreement on its standard and effective treatment method. Sabry and Al-Hendy (2012) state that apart from DPL, uterine leiomyomas, commonly known as fibroids, are the most common female condition globally [7]. The condition has the same characteristics as DPL and is more common in African American women. Just like DPL, uterine leiomyomas cause multiple reproductive issues such as pelvic pressure, excessive bleeding, miscarriage, and infertility. Doctors can treat the condition through surgery for better outcomes [7].

Furthermore, genetic factors correspondingly play a central role in the pathogenesis of peritoneal leiomyomatosis. A study by Izi et al. (2023) indicates that genetic alterations and chromosomal anomalies are related to leiomyomas, such as fluctuations in genes linked to cell series regulations, extracellular matrix model, and apoptosis [8]. Normally, these genetic changes contribute to flat muscle cells’ uncontrolled growth and survival, leading to leiomatosis development. Also, an experimental analysis by Vaishnavi et al. (2024) discovered the potential association between genetic disposition, hormonal disbalance, and iatrogenic features in the pathogenesis of peritoneal leiomyomatosis [9]. Liu et al. (2021) reveal the iatrogenic nature of DPL, demonstrating that morcellation during such as myomectomy or hysterectomy may result in the dissemination of leiomyomatous tissue within the peritoneal cavities, leading to the development of DPL [10]. 

#### Differentiating Uterine Leiomyomas from DPL

While uterine leiomyomas and DPL share histological similarities, there are critical differences in their presentation and potential for development following surgical intervention: uterine leiomyomas are typically confined to the uterus and present as well-defined, benign tumors. They can vary in size and number, often causing symptoms such as heavy menstrual bleeding, pelvic pain, and pressure effects on adjacent organs. When removed surgically, especially through morcellation, there is a potential risk for dissemination, but not all uterine leiomyomas develop into DPL. Proper surgical techniques and containment methods can mitigate this risk. Morgan et al. (2022) report on incidental findings of DPL and emphasize the importance of recognizing the potential for dissemination following surgical procedures, particularly with morcellation [5].

Meanwhile, DPL is characterized by multiple leiomyomas scattered throughout the peritoneal cavity, often mimicking malignant processes. DPL can be asymptomatic or cause nonspecific symptoms like abdominal pain and bloating. Hormonal influences, particularly estrogen, play a significant role in the development of DPL. Surgical procedures like morcellation during myomectomy can contribute to the spread of leiomyoma cells throughout the peritoneal cavity. Rosati et al. (2021) discuss cases of DPL with and without malignant transformation, underscoring the significance of surgical history and hormonal factors in the pathogenesis of DPL [2].

### 2.2. DPL Diagnosis

DPL may happen within the medicinal exhibition, extending from asymptomatic related results to symptomatic cases with substantial abdominal pain. The variance in medicinal exhibition largely depends on the size, location, and level of the peritoneal leiomyomatous nodules within the peritoneal cavities. Huang et al. (2021) argue that the symptomatic incidences of PL are chiefly categorized by abdominal pains, discomfort in the lower abdomen, and swelling [11]. Such signs might lead to the widespread impact of leiomyomatous nodules in the abdominal and pelvic cavities. Again, the patients might experience urinary symptoms, including frequent urination, irritation, and pain while urinating, significantly when the nodules affect or compress the urinary bladder. Moreover, several or large nodules can result in gastrointestinal symptoms, including bowel obstruction or constipation, by applying a lot of pressure on the intestines.

According to Huang et al. (2021), the diagnostic methods for DPL include computer tomography, ultrasound, magnetic resonance imaging (MRI), and histopathological assessment [11]. These procedures are central in determining the DPL diagnosis and evaluating its extent in the body. They also guide healthcare professionals during surgical interventions. Also, DPL can sometimes be discovered when physicians conduct surgical procedures for unconnected conditions. Since it takes a long time for the patient to feel the symptoms, DPL can be present in the body without showing any symptoms. According to Xu and Qian (2019), transvaginal three-dimensional sonography is a crucial imaging device for assessing pelvic pathology, such as leiomyomatous nodules within the peritoneal cavity [12]. The method enables healthcare professionals to have a comprehensive size visualization, location, and number of nodules, which helps in surgical planning, procedures, and management decisions.

Correspondingly, MRI is a vital imaging method for DPL characterization. It offers high and clear resolution images that show the spread and morphology of leiomyomatosis nodules, assisting the physicians in differentiating them from other pelvic masses, which can cause DPL. Again, MRI can similarly evaluate the nodules’ vascularity utilizing methods such as contrast-improved imaging, which is primarily helpful in sarcomatous changes or when deadly deterioration is suspected. A review by Yang et al. (2020) comparing different DPL diagnostic methods revealed that histopathological assessment remains the critical standard for determining DPL diagnosis [4]. The tissue samples are obtained from leiomyomatous nodules during surgical procedures such as laparoscopy or exploratory laparotomy, where they undergo extensive histological assessment. After the analysis, the presence of characteristic spindle-shaped smooth muscle cells and immunohistochemical staining series, comprising the positivity for specific smooth muscle markers such as SMA and desmin, indicates DPL diagnosis.

Ryu et al. (2020) study outcomes indicate the need to include numerous diagnostic approaches, such as MRI, transvaginal three-dimensional sonography and histopathological evaluation, to promote general assessment and management of DPL [13]. Such methods help to establish the correct diagnosis and guide during therapeutic interventions, including surgical nodule resection to improve patient outcomes. Most healthcare facilities use these techniques to ensure effective DPL diagnosis and treatment.

In differential diagnosis, DPL should be differentiated from other diseases with almost similar characteristics, such as uterine leiomyomas, pelvic masses, ovarian tumors, endometriosis, and peritoneal carcinomatosis [11]. The major similar pathologies are considered the following:○Malignant peritoneal mesothelioma that typically presents with diffuse thickening of the peritoneum, which can be distinguished from DPL by its pattern and associated calcifications. Histopathology is crucial for differentiation, as mesothelioma shows a diffuse malignant cell pattern, unlike the well-circumscribed leiomyomas seen in DPL [1].○Ovarian leiomyomatosis that typically manifests as localized ovarian masses rather than diffuse peritoneal lesions. Histologically, these are similar to uterine leiomyomas but confined to the ovaries [11].○Sarcomas (including leiomyosarcomas) that exhibit more aggressive features, including high cellularity, atypical mitoses, and necrosis. Imaging may show irregular masses with heterogeneous enhancement [12].○Other tumors such as gastrointestinal stromal tumors (GISTs) that are differentiated by immunohistochemical profiles (e.g., CD117 positive) and distinct cellular morphology.

Thus, clinicians should be careful when establishing the critical characteristics of DPL to avoid confusion. Different imaging outcomes help diagnose DPL effectively, resulting in suitable treatment and prevention methods.

## 3. Results

The scoping review of DPL reveals critical insights into this rare condition’s demographic, clinical presentation, and management. The data extracted from the literature presents a comprehensive overview that aids in understanding the disease’s impact and the effectiveness of current treatment strategies (Table 1).

### 3.1. Gestation/Para

The most common gestation/para status among the articles is women who have never been pregnant, followed by those with one pregnancy and one birth. The average gravidity (number of times a woman has been pregnant) is 1.85, with a range of 0–4 pregnancies. The average parity (number of times a woman has given birth) is 0.77, with a range of 0–2 births. From the individual cases, three cases mention women who have never been pregnant (nuligesta, nulliparous, gravida 0). Also, the women were described as G1P1, indicating one pregnancy and one birth.

Two cases describe women as para 2/0/0/2, which typically means two births.

One case describes a woman as P1L1, indicating one pregnancy and one living child.

One case describes a woman with an obstetric history of G3P2A11, which seems to be a typographical error but suggests three pregnancies.

### 3.2. Age

DPL is a condition that is most common among women of the middle-aged with the mean age at diagnosis being 38.5 years. It is a rising concern since this demographic targets women of child-bearing age; therefore, influence on fertility and gynecological health may be affected [14]. In terms of their reproductive potential, DPL poses significant consequences because the condition often requires treatments that may harm an individual’s ability to bear children or result in decisions about child-rearing.

### 3.3. Symptoms

The main clinical symptoms of DPL are abdominal and pelvic pain, which is rather vague and can manifest in several similar gynecological and other conditions [1,9,15,16]. This leads to the manifestation of symptoms that are similar to many other diseases, which in turn causes serious diagnostic confusion where such vague symptoms are attributed to common diseases [8]. Thus, DPL should be considered in differential diagnoses, especially when patients present with these symptoms and have no obvious etiology. Any modification or increase in clinician’s awareness and suspicion of DPL can be useful in identifying patients and initiating appropriate interventions earlier, which would benefit the patient.

### 3.4. Involvement Sites

The common sites involved in DPL are the peritoneum, the omentum, and the mesentery (Figure 2). This pattern of dissemination is in keeping with the disease’s ranging leiomyomatous nodules in different parts of the peritoneal cavity [6,13,17]. These nodules may have complications in relation to their size, number, and placement [18]. For example, nodules mainly enumerated in the peritoneum may result in marked bowel obstruction, while nodules arising from the omentum or mesentery may cause urinary signs and symptoms such as frequent urination, urgency, and displeasure. Therefore, in certain cases, the nodules produce a substantial increase in size and lead to mass effect or compression of the surrounding organs and tissues, changing the clinical picture and necessitating an adequate therapeutic management plan.

DPL usually requires surgical intervention due to high operative morbidity consisting of as many nodules as possible to relieve symptoms and avert severe consequences. Nevertheless, the decision to operate must be made after a prudent analysis of potential benefits concerning the costs and risks related to the surgical intervention, including the patient’s age, desire for fertility, and overall health condition. Thus, as researchers uncover more about DPL, it is expected that future treatment options to manage DPL and its effects will be more specialized and less invasive to patients.

### 3.5. Surgical Procedures

Laparotomy has been deemed the most prevalent surgical procedure for DPL treatment. This procedure is particularly helpful in ensuring the surgical team has an adequate view and access to the abdominal cavity, which is instrumental in the delicate task of resecting leiomyomatous nodules [19]. From a comprehensive approach, the main advantage of laparotomy includes sufficient possibilities to study the abdominal cavity, remove several localized nodules, and use other less invasive techniques (Figure 3).

Hysterectomy is usually conducted concurrently as a part of the surgical management plan in situations where the uterus is involved in the pathophysiology or in cases where fertility is not a priority for the patient [5]. This approach can be beneficial in the case of no recurrence of DPL, as the uterus can be a source of new multiple nodules.

Resection of nodules is a viable approach through DPL, which is intended to eventually relieve the patient’s complaints and prevent the occurrence of additional complications that may stem from further growth or spread of nodules [4]. The aim of this surgical intervention is not only the definitive treatment of the pathological conditions since the nodules cause voluminous masses that can obstruct the bowel the renal pelvis, or affect other organs and cause dysfunction.

#### 3.5.1. Number of Nodules

An average of ten nodules are removed during these procedures, though there is variance from one patient to another. These differences can be attributed to the median time from the disease onset to diagnosis, as this could establish the number and size of the nodules in question [12]. Additionally, hormonal factors may play a role in influencing the growth rate of leiomyomatous nodules, while genetic factors may help elucidate variations in disease manifestation and treatment outcomes.

Regarding the surgical management of DPL, there are several factors that can be taken into consideration regarding the process. The changes in nodule number, the nature of DPL development, and the skepticism for a single monogenic cause all contribute to the need for personalized treatment and enhancement of patient management. Therapeutically, operation has continued to be considered the optimal intervention for this disorder, and more specifically, laparotomy enhances symptomatic relief with a lowered risk of complications.

#### 3.5.2. Follow-Up Period

The follow-up time for patients who have been treated for DPL ranges from 3 to 49 months, which underlines a need for long-term monitoring. This is important because the majority of patients who undergo surgical treatment of the condition stay free of symptoms for an extended post-surgery period, implying that surgical interventions help in managing the condition. But what really keeps people alert is the unexpected return of the disease or the appearance of new nodules even when the individual does not show any sign of discomfort.

These results show the variability in the follow-up and, therefore, indicate that DPL is unpredictable and could result in recurrence at a later date; hence, constant follow-up should be observed. Imaging methods, including ultrasonic, computed tomography, or magnetic resonance imaging, serve as an integral part of this process, allowing the visualization of any pathological changes within the abdominal cavity at early stages [20]. It has been established that early detection of new or recurring nodules is especially important in order to start a further surgical intervention or utilize other strategies.

Consequently, the prescription of a directed follow-up program can help monitor the health status of the patients treated through DPL and their general well-being. It helps in eradicating any complication that might arise and promotes the quality of life of the patient in the future. The adherence to the follow-up indicates a constructive strategy in the delivery of patient care, which focuses on consistent care rather than a one-shot cure solution.

#### 3.5.3. DPL Management

DPL management entails an all-inclusive approach considering several factors, such as patients’ signs and symptoms, reproductive needs, lesion traits, and possible risks of malignant modification. Xu and Qian (2019) state that the main purpose of DPL management is to lessen its risk features, reduce symptoms, prevent disease advancement, and diminish the risk of problems, such as sarcomatous variations and nodule distortion [12]. Healthcare amenities have implemented contemporary approaches to manage DPL to guarantee enhanced patient results. A study by Chen et al., 2021) revealed that surgical intervention raised concerning the application of morcellation during the laparoscopic procedures, which can possibly result in the spread of leiomyomatous tissue [19].

Healthcare professionals use open surgical or laparoscopic methods depending on the size of the lesion, location, and patient factors. The laparoscopic resection is most common because of its marginally painful and complex nature and quick recovery time [21]. The patient takes one to two weeks to fully recover from the procedure. However, some issues have been raised concerning the application of morcellation during laparoscopic procedures, which can potentially spread leiomyomatous tissue and increase the risk of DPL recurrence [9]. Therefore, there have been advancements in surgical procedures, mainly emphasizing tumor removal without morcellation. Surgical procedures have significantly reduced the cases of DPL reoccurrence or spread.

According to Vimercati et al. (2022), there is a need to adopt alternative methods that are comprehensive and better than morcellation in laparoscopic surgery to lessen the risk of DPL growth and spread [18]. Methods such as manual rotary lesion cuts in a fetch bag or nodule removal using a posterior fornix incision have been proposed as more effective and safer for tumor removal without morcellation. The innovative approaches decrease the DPL risk of its spread, shorten the process span, reduce expenses, and improve patient results.

Further, conservative DPL management is useful when the patients are asymptomatic or when DPL is steady without substantial growth or problems. The approach involves regular monitoring of lesion signs and size with imaging tools such as MRI ultrasound or medicinal examination [17]. Hormonal treatments such as gonadotropin-releasing hormone aromatase or agonist inhibitors are applied to manage DPL and its therapies to reduce leiomyomatous growth and spread and lessen symptoms, particularly in premenopausal females with estrogen-dependent abrasions.

Medical or surgical innovations, including applying fetch bags and physical rotary cut approaches during laparoscopy, have significantly become more effective DPL management approaches because they aid in lessening the risk of DPL dissemination and improve surgical outcomes [14]. Surgical inventions, together with exhaustive surgical scheduling and adherence to tumor-free procedures, meaningfully enhance patient care and reduce reoccurrence.

Consequently, DPL management normally involves a multidisciplinary team, including radiologists, pathologists, gynecologists, and oncologists, to promise a wide-ranging examination, accurate diagnosis, appropriate treatment arrangement, and excellent care delivery to the patients. Overall combined decision-making processes from such professionals grounded on illness traits and patient influences are crucial to achieving promising outcomes and decreasing potential risks connected with DPL management methods. Gynecologists, in this case, serve a fundamental role in DPL diagnosis, assessing its impact on a patient’s reproductive health and recommending appropriate treatment approaches such as surgeries and hormonal treatments [16]. Further, oncologists are very important in case DPL displays symptoms of lethal transformation or if there are cancer risk concerns and to guarantee widespread examination, treatment, and management.

Correspondingly, radiologists assist in image study and interpretation using CT scans, MRI, and ultrasound, providing detailed and correct understandings of lesion features, participation period, and probable problems. Pathologists commonly examine the biopsy samples obtained during a surgical procedure to determine the DPL diagnosis, assess historical features, and identify any warning signs of sarcomatous variations, as well as direct prevention and treatment decisions and predictive evaluation for enhanced patient outcomes [9]. Similarly, multidisciplinary teams carry out an inclusive assessment of the patients, considering features such as signs, lesion size, setting, reproductive purposes, age, and overall health status to develop the most operative strategies for DPL management.

The team similarly gives efficient patient education and counseling on managing DPL. Some of the information received by patients concerning DPL comprises its signs, etiology, treatment approaches, diagnostic procedures, prevention techniques, probable risks, and projected outcomes. The information assists patients in making well-informed decisions that align with their standards and needs and assists them in managing DPL more efficiently [12]. Accordingly, DPL can be managed through clinicians’ monitoring and follow-ups. Most clinicians, after treating DPL patients, start regular follow-up appointments to assess the treatment efficiency, monitor postoperative recovery, and notice any reoccurrence or development of DPL, which promotes operative DPL management [15]. Healthcare professionals similarly use imaging research, such as MRI, CT scans, and ultrasound, to assess the patient’s response to treatments and establish any new lesions or modifications of the normal ones.

## 4. Discussion

The scoping review of DPL highlights some complexities of this rare disease most prevalent among middle-aged women. The general patient population has non-specific symptoms like abdominal discomfort and pelvic pain, which makes diagnostic procedures complicated [11]. Such symptoms, together with the frequented areas of involvement, such as the peritoneum, omentum, and mesentery, make it crucial for clinicians to have heightened suspicion.

Surgical management, especially laparotomy, has remained the primary treatment model for DPL, as seen by the general consensus among surgeons. This is because there is a need to mitigate the symptoms and possibly avoid complications arising from the development and progression of leiomyomatous nodules [21]. Nevertheless, the distribution of nodules, which was approximately ten, with some differences among patients and a variance in follow-up period ranging from 3 to 49 months, also puts forward the need for an individualized treatment plan. Any such plans have to take into account specific features of the patient, period of the disease, hormonal background, and possibly genetic predispositions.

The post-treatment phase is also the most informative, as most patients will not experience further symptoms, reinforcing the effectiveness of the surgical procedures performed. However, it also raises questions about the likelihood of the presence of some other signs of disease development that may have remained unnoticed [10]. This concern emphasizes the need for more extensive research on DPL and other non-operative treatments, as well as long-term prognosis studies. It is also relevant to discuss the influence of hormonal and genetic processes in disease development and relapse, as well as indicate the potential for further research.

DPL, therefore, challenges the medical fraternity for more rigorous diagnosis and control. Therefore, the findings from this review can help to improve clinical practice and assist physicians in identifying the condition and applying the proper management interventions at the right stage. Therefore, additional studies should be conducted with the aim of improving the diagnostic classification, identifying less invasive treatment strategies, and understanding the favorable or unfavorable outcomes of DPL patients. These efforts are essential in increasing the quality of patient care and the well-being of individuals with this challenging disorder.

## 5. Conclusions

DPL is an uncommon condition characterized by multiple leiomyomas in the pelvic cavity and lower abdomen, predominantly affecting women. With fewer than 200 cases reported worldwide, most of which involve women of reproductive age, DPL remains a rare diagnosis. A significant iatrogenic factor contributing to DPL is surgical procedures, particularly morcellation during myomectomy. Effective diagnosis typically involves computed tomography, ultrasound, magnetic resonance imaging (MRI), and histopathological evaluation. Management of DPL requires a multifaceted approach, including surgical intervention with a focus on minimizing iatrogenic risks, patient education, counseling, regular follow-ups, and conservative management when appropriate. Preventive strategies during surgery, such as avoiding morcellation and adopting refined techniques, such as endobag morcellation, are crucial to reducing iatrogenic dissemination. Collaboration within a multidisciplinary team can further enhance the management and outcomes for patients with DPL. Addressing these iatrogenic risks and improving surgical practices are essential for better prevention and management of this rare condition.

## Figures and Tables

**Figure 1 biomedicines-12-01749-f001:**
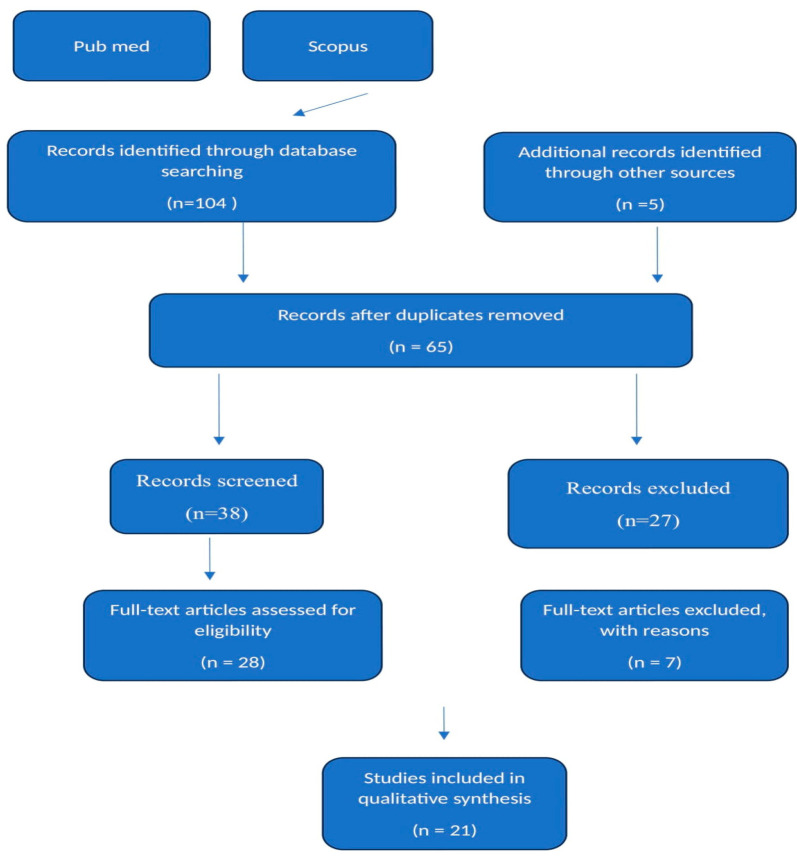
Scoping review flow diagram. Caption: the PRISMA flow diagram for the scoping review detailing the database searches, the number of abstracts screened, and the full texts retrieved.

**Figure 2 biomedicines-12-01749-f002:**
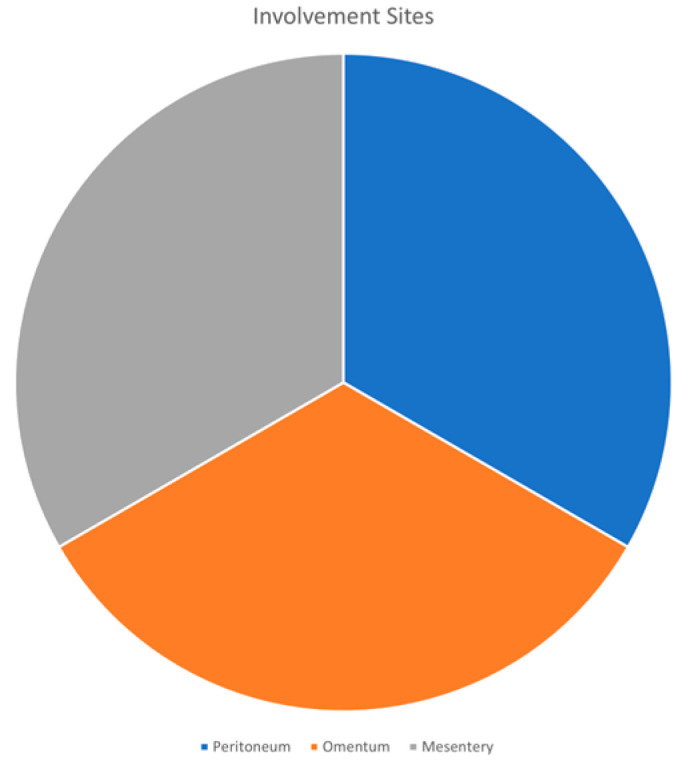
Involvement sites of DPL.

**Figure 3 biomedicines-12-01749-f003:**
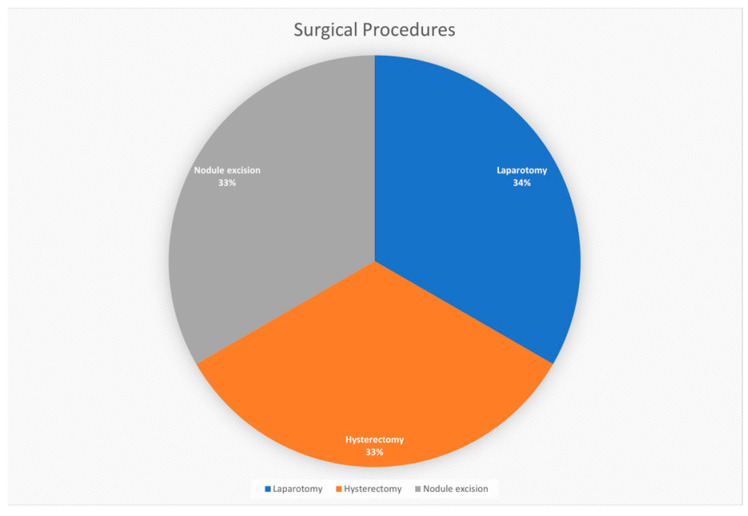
Types of surgical procedures used in DPL.

**Table 1 biomedicines-12-01749-t001:** Patient characteristics from the selected articles.

Characteristics	Data
Gestation/para	Nullipara/1/2
Average age	38.5 years
Most common symptoms	Abdominal discomfort, pelvic pain
Frequent Involvement sites	Peritoneum, omentum, mesentery
Types of surgical procedures	Laparotomy, hysterectomy, nodule excision
Average number of nodules	10
Range of follow-up periods	3–49 months

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
