# Peer review of "Disseminated Peritoneal Leiomyomatosis—A Challenging Diagnosis-Mimicking Malignancy Scoping Review of the Last 14 Years"

_biomedicines, 2024, doi:10.3390/biomedicines12081749_

Round 1

Reviewer 1 Report

Comments and Suggestions for Authors

The topic of disseminated peritoneal leiomyomatosis is not novel. One of the primary causes of this disease is iatrogenic, mainly due to surgical procedures, which has been mentioned in multiple literatures. I believe the author should emphasize this primary cause in the abstract and conclusion, and discuss how to better prevent iatrogenic causes during the surgical removal of uterine leiomyomas. Additionally, there should be a detailed explanation on how to differentiate it from other similar tumors. Furthermore, the differences between uterine leiomyomas and disseminated peritoneal leiomyomatosis should be discussed, as some uterine leiomyomas do not develop into disseminated peritoneal leiomyomatosis after morcellation of uterine leiomyomas.

Comments on the Quality of English Language

There is a language issue in the text. In the fifth paragraph of the Materials and Methods section, where it mentions the infection rate of DPL, I believe the author wants to write the incidence rate of DPL.

Reviewer 2 Report

Comments and Suggestions for Authors

In my opinion, a very interesting manuscript. The authors have addressed the interesting topic of  disseminated peritoneal leiomyomatosis

In the assessed manuscript definitely lacks a flowchart that can be used to document the sources you used in your review, and your selection and screening processes. Please include a flowchart in this manuscript consistent with the PRISMA recommendations.

In my opinion, it is worth improving the section “ conclusions”  - is written too laconically, while it should highlight the merits of the analysis made by the authors during the editing of this manuscript

Once the proposed revisions have been made, the manuscript can be published

Round 2

Reviewer 1 Report

Comments and Suggestions for Authors

I agree to publish this article.